

Earth System
Science
Data

# Continuous national gross domestic product (GDP) time series for 195 countries: past observations (1850–2005) harmonized with future projections according to the Shared Socio-economic Pathways (2006–2100)

**Tobias Geiger**

Potsdam Institute for Climate Impact Research, Telegraphenberg A 56, 14473 Potsdam, Germany

**Correspondence:** Tobias Geiger (geiger@pik-potsdam.de)

**Abstract.** Gross domestic product (GDP) represents a widely used metric to compare economic development across time and space. GDP estimates have been routinely assembled only since the beginning of the second half of the 20th century, making comparisons with prior periods cumbersome or even impossible. In recent years various efforts have been put forward to re-estimate national GDP for specific years in the past centuries and even millennia, providing new insights into past economic development on a snapshot basis. In order to make this wealth of data utilizable across research disciplines, we here present a first continuous and consistent data set of GDP time series for 195 countries from 1850 to 2009, based mainly on data from the Maddison Project and other population and GDP sources. The GDP data are consistent with Penn World Tables v8.1 and future GDP projections from the Shared Socio-economic Pathways (SSPs), and are freely available at http://doi.org/10.5880/pik.2017.003 (Geiger and Frieler, 2017). To ease usability, we additionally provide GDP per capita data and further supplementary and data description files in the online archive. We utilize various methods to handle missing data and discuss the advantages and limitations of our methodology. Despite known shortcomings this data set provides valuable input, e.g., for climate impact research, in order to consistently analyze economic impacts from pre-industrial times to the future.

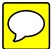

## 1 Introduction

The concept of measuring and comparing economic activity within and across countries using the gross domestic product (GDP) is rather new in historic terms. Starting with some first attempts to quantify economic activity in the late 19th century for certain countries, comprehensive regular assessments of GDP were only established in the second half of the 20th century. Since then GDP has become the standard indicator to assess nations' development, despite initial and more recent criticism concerning the incomplete representation of a nation's state via GDP and the potential problematic comparison across countries (Speich Chasse, 2013).

Nonetheless and because of a lack of alternatives GDP has proven to be a useful measure to track the evolution of economic development within or across nations. Many other development proxies, e.g., the level of education, life expectancy, the population's health status, and many others, have been shown to correlate well with a nation's GDP; see, e.g., Gennaioli and La Porta (2013). Similarly, a reduction in vulnerability (or an increase in resilience) to natural disasters has also been shown to correlate well with a nation's GDP (Kousky, 2013), resulting in less mortality and in fewer damages relative to GDP. Most research in this field focuses only on the last decades where sufficient coverage of economic activity exists for most countries of the world. However, many fields of research could benefit from a more comprehensive economic data set that covers a larger time horizon and a larger number of nations to gain a better understanding of the drivers of long-term economic development.

Various global institutions (e.g., Worldbank, Organisation for Economic Co-operation and Development, OECD; International Monetary Fund, IMF) and research groups (e.g., Penn World Table) have therefore assembled global GDP data sets, most of which provide a comprehensive view across space but lack data prior to the 1960s. In addition, there have been attempts to provide GDP or income (i.e., GDP per capita) estimates reaching further back in time based on proxies for specific periods; see, e.g., Baier et al. (2002), Mitchel (2003), Maddison (2007), and Bolt and van Zanden (2014). However, these estimates only provide snapshots of economic activities for specific periods without continuous global coverage across time.

Here, we contribute to increasing the usability of historical economic time series by creating a continuous and complete income and GDP time series from 1850 to the present. We do so by combination of various data sources and methods to interpolate and extrapolate missing data points in a pragmatic but most sensible way. The Maddison Project Database (MPD) thereby constitutes the foundation of the period mostly before 1960, while the Penn World Table (version 8.1) sets the basis for the more recent past. Our final GDP time series covers 195 countries (in their present constitution) from 1850 to 2009 and is consistent with GDP projections from the Shared Socio-economic Pathways (SSPs) that extend the historical time series from 2010 to 2100.

The long record and complete coverage enhance the data set's usability. It has already been assigned as input data for the current climate change impact model runs within the global Inter-sectoral Impact Model Intercomparison Project (ISIMIP2b, www.isimip.org) (Frieler et al., 2017), and has been used in a downscaling approach to provide spatially explicit economic information on the grid level (Geiger et al., 2017; Murakami and Yamagata, 2017) that can, e.g., be used to quantify economic values exposed to climate extremes (Geiger et al., 2018). Despite various known shortcomings that are discussed in detail below, this new data set has broadened and will further broaden the applicability of historic estimates of economic activity and potentially feed back to foster increased research interest in the field of economic history and the improvement of the current data set.

## 2 Data and methods

In the following we present our methodology that is used to create a continuous and consistent GDP time series for 195 countries based on national accounts data from the Penn World Table (PWT), the MPD, World Development Indicators (WDI), the History Database of the Global Environment (HYDE), and projections from the SSPs. We will present the data sources first before we describe the consistent merging procedure across all sources, as additionally summarized in Table 1.

**Table 1.** Overview of data sources used to create the final data product ranked according to their priority. Please refer to Sects. 2.1 and 2.2 for details on the data sources.

| Data priority | Historical observations | | Future projections | |
|---|---|---|---|---|
| | Income | Population | Income | Population |
| 1 | PWT8.1 | PWT8.1 | OECD SSP2 | |
| 2 | MPD | HYDE | – | – |
| 3 | PWT9.0 | PWT9.0 | – | – |
| 4 | WDI | WDI | – | – |

### 2.1 Gross domestic product (GDP)

#### 2.1.1 Penn World Tables (PWTs)

The PWTs comprise national accounts data maintained by scholars at the University of California and the University of Groningen to measure real GDP across countries and over time. The database is successively updated and extended, with the latest release being PWT 9.0 (Feenstra et al., 2015), and provides the most extensive coverage for GDP reported in purchasing power parity (PPP) across time. We here mostly rely on the PWT release 8.1 from 2015 for two reasons: first, PWT8.1 data are reported in 2005 PPP USD and are thus consistent with SSP projections. Second, PWT8.1 is in close agreement with the SSP initial data in 2010, thus reducing matching artefacts to a minimum. Moreover, PWT8.1 replaces the strongly criticized original PPPs for 2005; see, e.g., Deaton and Heston (2010), by a modified version; see Inklaar and Rao (2017) for details. Missing countries in PWT8.1 are taken from PWT9.0 after rescaling from 2011 to 2005 PPP USD; see the discussion below. PWT also provides national population estimates that we apply to generate income estimates based on national GDP.

#### 2.1.2 World Development Indicators (WDI)

The WDI assembled by the Worldbank provide a vast resource of socio-economic data. Their present release of PPP-based GDP comes in 2011 PPP USD and is available for 1990 to 2015 (Worldbank, 2017). As income estimates in 2005 PPP USD values are no longer available, we rescale from 2011 PPP USD to 2005 PPP USD to insert otherwise missing countries in the PWT data; see the discussion below. The WDI also provide national population estimates that we apply to generate income estimates based on national GDP.

#### 2.1.3 Shared Socio-economic Pathways (SSPs)

The SSPs are storylines of plausible alternative evolutions of society at the global level that can be combined with assumptions about climate change and policy responses to evaluate climate change impacts, adaptation, and mitigation

Please note the remarks at the end of the manuscript.

(O'Neill et al., 2017). Meanwhile different integrated assessment models (IAMs) have generated GDP projections along the SSP storylines. The associated national time series all start in 2010. The historical data set we provide is designed to allow for a smooth transition from historical time series to the associated future projections. As such it can, e.g., be used for transient climate impact model simulations covering the historical period and future projections under different socio-economic development pathways as described by the SSPs.

SSP projections are generally available at https://tntcat.iiasa.ac.at/SspDb/dsd?Action=htmlpage&page=about. We here use the OECD GDP data set (Dellink et al., 2017) provided by the Inter-Sectoral Impact Model Intercomparison Project (ISIMIP), freely available at https://www.isimip.org/gettingstarted/availability-input-data-isimip2b/ upon registration. This data set was directly provided by the OECD and is advantageous as it contains data on seven additional countries. Note that one of these additional countries (Aruba, ABW) was excluded due to data issues. (The reported SSP GDP value for 2010 is a factor of 10 smaller than observational records.)

### 2.1.4 Maddison Project

The Maddison Project resembles a cooperative effort to assemble historical national accounts (Bolt and van Zanden, 2014), continuing the ground-breaking work by the late Angus Maddison. The MPD is a freely available Excel spreadsheet (Maddison, 2016) that provides per capita income (in 1990 Geary–Khamis, G–K, dollars) for 184 countries and/or world regions between 1AD and 2010 in varying time intervals. Since 1800 data have been provided annually, containing however large fractions of missing values, in particular for the African continent, Western Asia and the former Soviet Union. The global missing value fraction is 61.8 % between 1800 and 2010. It steadily decreases to 51.7 % (36.9 %) when analyzing the data from 1850 (1900) onwards. After 1950, when national accounts data were starting to be routinely collected, the missing value fraction drops to 8 %. The data are nearly complete since 1990 (missing value fraction: 3.6 %). See Fig. 1 and the supplementary data in the DOI data archive (Geiger and Frieler, 2017).

We selected 1850 as the start year for our present work for two reasons: first, coverage is somehow better than in 1800, and second, we can rely on available data points between 1800 and 1849 for the purpose of interpolation rather than extrapolation.

### 2.2 Population data

Parts of the GDP data sets (PWT and WDI) also provide national population data to derive income. Whenever available we use the associated population estimates from the same source to derive income from GDP. However, to estimate national GDP from the income data generated within the MPD we use population estimates from the HYDE data set.

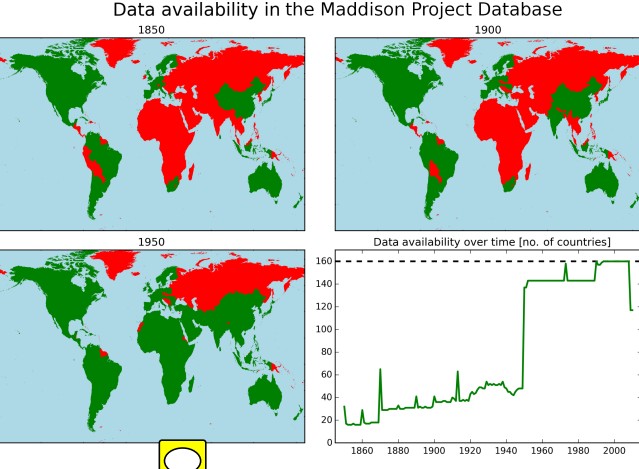

**Figure 1.** TS4 Illustration of data availability in the Maddison Project Database CE3 over time. Maps show the geographical distribution of available data points for 3 selected years (1850, 1900, 1950), while the plot displays the actual number of available countries over the full period. Countries are displayed using their current borders. We only show data for those 160 countries that are explicitly named in the Maddison Project Database; country group estimates lacking national resolution are not shown (see Table 2 for details).

### History Database of the Global Environment (HYDE)

HYDE is developed under the authority of the Netherlands Environmental Assessment Agency and provides (gridded) time series of population and land use for the last 12 000 years (Klein Goldewijk et al., 2010, 2011). HYDE provides national population data decennially up to 2000 and annually up to 2015. Where required we linearly interpolate the data to derive annual distributions.

### 2.3 Completing the Maddison Project Database

The MPD provides income data on a national and supranational level, where the supranational data represent population-weighted averages of national values (see Table 2). In addition, there exist three world region-specific groups of small countries (groups of 14 small European countries, 21 Caribbean countries, and 24 South-East Asian countries) which lack national resolution; i.e., member countries are prescribed identical growth paths (see Table 2). In the following, each of the three country groups is treated like an individual country with respect to replacement of missing data.

Generally, gap filling is first done on the supranational level and later gaps in national time series are filled by accounting for growth rates of either neighboring countries, the

Please note the remarks at the end of the manuscript.

**Table 2.** Overview of the world region-specific supranational country groups and the small country groups that lack national resolution, as used within the Maddison Project Database.

| World region | Supranational group 1 | Supranational group 2 | Small country groups |
|---|---|---|---|
| Europe | 12 Western European (WE) | 7 Eastern European (EE) | 14 small European countries |
| Latin America | 8 Latin American (LA) | – | 21 Caribbean countries |
| Asia | 16 Eastern Asian (EA) | Western Asian (WA) | 24 South-East Asian countries |
| Africa | African total (AT) | – | – |

associated supranational level, or the associated world region the country belongs to.

In the following we present our gap-filling methodology, first general steps and then for each world region in detail, starting with regions with the least missing values.

### 2.3.1   Preparatory steps

As a first step we populate all missing data points in 1850, the initial year of our data product, by linear interpolation between the last available data point before 1850 and the first one after 1850, ensuring that it is not more distant in time than 1870. Next, and if available, we generate annual data by linear interpolation of data points between 1850, 1860, and 1870. These preparatory steps reduce the missing value fraction from 51.7 to 48.5 %.

### 2.3.2   Europe

The preparatory steps completed the country-level data for most countries in Western Europe. These individual country-level data are then used to complete the time series for the supranational group of 12 Western European (WE) countries by population-weighted mean income using HYDE's national population data, as growth rates from the WE time series are used to approximate missing values for the country group of 14 small European countries. Similarly, the United Kingdom's growth path is used to complete Ireland's time series.

Next, gaps in the supranational group of seven Eastern Europe (EE) countries time series are linearly interpolated (including the time of World War 2) and then, starting in 1870, used to extrapolate individual country growth paths in Eastern Europe back to 1850.

Relative changes in the former Yugoslavia's time series are used to extrapolate their temporary constituents back to 1850, and to extrapolate Kosovo's income between 1991 and 2010. The same procedure is conducted for the constituents of the former Czechoslovakia.

Relative EE changes prior to 1885 are used to complete the former USSR time series. The relative changes in the USSR time series are used to extrapolate income for all member countries of the USSR up to 1973, and to interpolate between 1973 and 1990. The fraction of missing values in the entire data set thus decreases to 34.4 %.

### 2.3.3   North America, Australia, and New Zealand

The preparatory steps were sufficient to fill the respective time series completely.

### 2.3.4   Latin America

Following the country grouping in the MPD, we generate a complete population-weighted average income time series for a group of eight large Latin American (LA) countries. To do so, Peru and Mexico's income is respectively extrapolated before 1870 and interpolated after 1870 based on the mean growth of the seven remaining countries. Relative changes in the LA time series are applied to estimate the income of the remaining South American countries. The time series for the group of 21 Caribbean countries is linearly interpolated and then used to fill gaps in the remaining Caribbean and Central American countries, except for Jamaica and Cuba, whose time series is sufficiently dense to be interpolated individually.

Finally, the fraction of missing values is reduced to 29.7 %.

### 2.3.5   Asia

Asia is separated into East and West Asia, reduced by the fraction that belonged to the former USSR.

Where required relative income changes based on the linearly interpolated time series for a group of 16 Eastern Asian (EA) countries are used to extrapolate individual country data before 1870. Furthermore, data gaps for selected countries with quite dense coverage (India, Japan, Indonesia (Java before 1880), Sri Lanka, and China) are filled by linear interpolation on an individual basis. Using a step-wise procedure, we fill the EA time series after 1870 with the population-weighted average income of all countries for which original data are available at any given time step. Relative changes in the EA time series are then used to fill gaps in all remaining Eastern and South-East Asian countries, in particular to complete the time series for the joint group of 24 South-East Asian countries. However, missing values for Bangladesh and Pakistan are estimated based on India's relative changes. Thereby, the fraction of missing values is reduced to 24.3 %.

In a second step, Western Asian countries including the Gulf States are treated, where data availability is very limited before 1950. Except for Turkey (former Ottoman Em-

pire, complete data since 1923), relevant data points exist only for 1820, 1870, and 1913. To also include countries with completely missing data before 1950, we assume that the income of Israel is equal to West Bank/Gaza income in 1870 and 1913, and that the income of some Gulf States is equal to Iran's income. For each country we linearly interpolate the data until 1913, and again until 1950, except for Kuwait, Qatar, and the UAE. For those three countries we estimate income growth to 1939 based on average income growth for all Western Asian (WA) countries and then linearly interpolate each country individually between 1940 and 1950, thereby ensuring that sharp income rises only occur after the discovery of oil.

Upon completing Asian time series the fraction of missing values is reduced to 19.5 %.

### 2.3.6 Africa

The MPD contains only six countries with income data prior to 1950: Egypt, Tunisia, Morocco, Algeria, and South Africa/Cape Colony (all since 1820), and Ghana (since 1870). Therefore, the African total (AT) population-weighted average income prior to 1950 is only defined by six countries. For historic and geographic reasons, we assume that those countries define the upper income limit when extrapolating the remaining countries back in time. So if a country's 1950 income is smaller than the six-country population-weighted mean, the income fraction is used to define the scaled income in 1913; otherwise, it is set equal to the six-country mean in 1913. Missing data between 1913 and 1950 are then interpolated according to scaled AT relative changes, while the bare AT relative changes are used to extrapolate missing values from 1913 to 1850.

### 2.3.7 Summary

Using all the steps described above, all missing values are filled and a complete time series (1850–2010) is obtained for all countries listed within the MPD. This time series, reported in 1990 G–K dollars and matched with IS03 country codes, can be found in the DOI data archive (Geiger and Frieler, 2017). Note that for some countries the reporting period ends in 2008; no extrapolation was done thereafter as more reliable data from other sources are available for this period.

### 2.4 Matching and time series conversion

Our final database is reported in PPP dollars referenced to 2005, the original unit of PWT8.1 and the SSPs. Consequently, data from the MPD, the WDI, and PWT9.0 require conversion. As for the MPD, no official conversion factors are available; time series of historic income from the MPD are scaled to systematically match PWT8.1 income data at the earliest reporting year of PWT8.1 data. This ensures that (1) time series for a large fraction of countries (18.5 %) with

no country-specific resolution in the MPD are assigned individual growth paths as soon these data are available, (2) data already provided in the finally desired currency unit (2005 PPP USD) are used preferably, and (3) the risk of overestimation of values in the distant past is reduced because of rising conversion factors with time (see Fig. 3 below for a more detailed discussion). The income in 2005 PPP USD is determined from the income in 1990 GK USD by using the country-specific conversion factor (CF) for the respective base year:

$$\text{Income (2005 PPP USD)}$$
$$= \text{Income}\,(1990\ \text{GK USD}) \cdot \text{CF}\,(\text{base year})\,.$$

To test for robustness the conversion factors are calculated for not only the earliest reporting year of PWT8.1, but also for each of the first 5 years. If the five individual conversion factors significantly vary (fraction of standard deviation of the first five conversion factors and the first conversion factor larger than the iteratively derived threshold of 4 %), we use the 5-year mean conversion factor instead. In total, 78 out of 195 countries require a mean conversion factor, while overlapping time series for 14 countries are so short that a mean conversion factor cannot be determined. All conversion factors and the base year of matching are available in the Supplement as well as supplementary data in the DOI archive (Geiger and Frieler, 2017). Figure 2 provides an illustration of the conversion factors, here shown as conversion rates in percent to highlight the symmetrical scattering around the cross-country median, which is larger than unity, as expected, due to the transformation forward in time between 1990 and 2005. The statistical distribution of conversion rates is shown for all 195 countries (left boxplot in Fig. 2), all 36 countries that do not appear specifically in the MPD but as country groups only ("no Maddison" boxplot in Fig. 2; see also Table 2), and separately for six different world regions. The outliers ARE (United Arab Emirates), BRN (Brunei), and SOM (Somalia) are explicitly stated in Fig. 2, making their respective converted time series before 1970 and 2010 rather uncertain. A large fraction of relatively large conversion rates stems from those countries with no individual resolution in the MPD ("no Maddison" boxplot in Fig. 2) that are matched to PWT8.1 data using the respective country group data; see Table 2. In contrast to the other world regions, Europe and South America show rather small variations in conversion rates that are in line with the relatively dense data coverage within the MPD.

Additional countries that are reported in the SSPs but not in PWT8.1 are included based on data first from PWT9.0 and second from WDI, except for El Salvador (SLV) and Zimbabwe (ZWE), where PWT9.0 data are chosen in place of the available PWT8.1 due to known issues in the data set; see https://www.rug.nl/ggdc/docs/what_is_new_in_pwt_81.pdf for details. Conversion of PWT9.0 data between 2011 PPP USD and 2005 PPP USD values is conducted by us-

Distribution of conversion rates

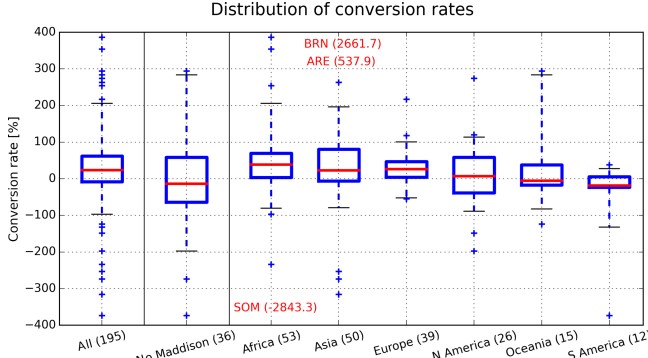

**Figure 2.** Statistical distribution of conversion rates used to convert 1990 GK USD to 2005 PPP USD. Most countries are close to the cross-country median conversion rate (+23.3 %, red line in the left boxplot), while SOM (Somalia), ARE (United Arab Emirates), and BRN (Brunei) (stated in red; ARE, BRN, and SOM also appear in the "all" boxplot, ARE additionally in the "no Maddison" boxplot) are clear outliers with respective conversion rates shown in parentheses. Whiskers display the 5 to 95 % percentile range. Numbers in parentheses on the horizontal axis indicate the number of countries included in the respective boxplot.

ing the PWT-provided exchange rates (PWT abbreviation: xr) and price levels of GDP (PWT abbreviation: pl_gdpo) to yield the following PWT- and country-specific conversion factor $CF_{PWT}$:

$$CF_{PWT}(2011\ \text{PPP USD} \rightarrow 2005\ \text{PPP USD})$$
$$= \left[ xr(2005) \cdot pl\_gdpo(2005) \right] / \left[ xr(2011) \cdot pl\_gdpo(2011) \right].$$

For WDI data we use the provided PPP conversion table (Worldbank, 2017) and corrections for inflation based on the USA GDP deflator (Worldbank, 2017) to yield the following WDI- and country-specific conversion factor $CF_{WDI}$:

$$CF_{WDI}(2011\ \text{PPP USD} \rightarrow 2005\text{PPP USD})$$
$$= PPP_{conv}(2005) / PPP_{conv}(2011) \cdot GDP_{defl}(2011 \rightarrow 2005),$$

where $GDP_{defl} \sim 0.89$ accounts for the price increase in the USA economy of about 12 % between 2005 and 2011. Where sufficient information was missing to convert between 2011 and 2005 PPP USD, we either used the USA GDP deflator only (for SLV, El Salvador, and ZWE, Zimbabwe), or used the conversion of a neighboring country (NRU, Nauru, instead of KIR, Kiribati). North Korea (ISO3: PRK) is only available in the MPD and is therefore excluded from the analysis.

In order to be consistent with SSP projections starting in 2010, we truncate the observation-based time series in 2005 and linearly interpolate from 2006 to 2009. Despite the fact that we drop 4 years of observational data including the unanticipated changes due to the global financial crisis (SSP products were produced many years prior to the financial crisis), we opted for this smooth procedure consistent with other

Relative change in 5-year mean conversion factors

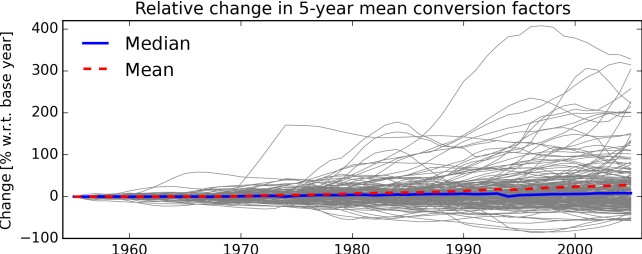

**Figure 3.** Base year dependent relative change in 5-year running-mean conversion factors for all 181 countries (gray lines) for which a 5-year mean can be determined. The cross-country mean (red-dashed) and the median (blue-solid) conversion factor increase by 28.2 and 8.4 % between 1954 and 2005, respectively.

SSP-informed data sources to avoid large unphysical kinks in the time series. Users that require more recent observational data are advised to rely on the underlying data sources directly. An illustrative example of the matching procedure for four selected countries is displayed in Fig. 4; the matching result for all 195 countries is shown in Fig. S1 in the Supplement.

As mentioned above, there are several reasons why we use the earliest year of data availability for PWT8.1 data as a base year for conversion, one of which being insufficient coverage for many countries in the MPD. While this is a sensible decision for most countries where conversion factors do not fluctuate rapidly over time, it can lead to distortions for some countries as, e.g., hyperinflation, economic crisis, or rapid economic growth may produce different conversion factors in different base years. In general, using earlier years for matching results in more conservative estimates of past income because mean and median conversion factors across countries grow with time; see Fig. 3.

Figure 3 also illustrates that changes in country-specific conversion factors over time cluster at relatively low values, while some countries show larger deviations and fluctuations from the mean development. Table 3 therefore provides a more detailed sensitivity analysis for 181 countries for which at least 5 years of overlapping data exist. For more than 70 % of the countries the change in conversion factors is smaller than 50 %, while almost 14 % of the countries show changes larger than 100 %. This sensitivity check provides useful information about the uncertainty of past income estimates: users should keep in mind that income estimates are inherently uncertain not only due to reporting issues and lack of data, but also due to currency conversion and their respective change over time. Country-specific conversion factors over time are provided for selected countries in Fig. 4 and for all countries in Fig. S1.

In a next step we multiply income and population time series to obtain GDP time series. As for the income matching between historical observations and SSP projections, population estimates are truncated in 2005 and linearly interpo-

## Matching results for selected countries

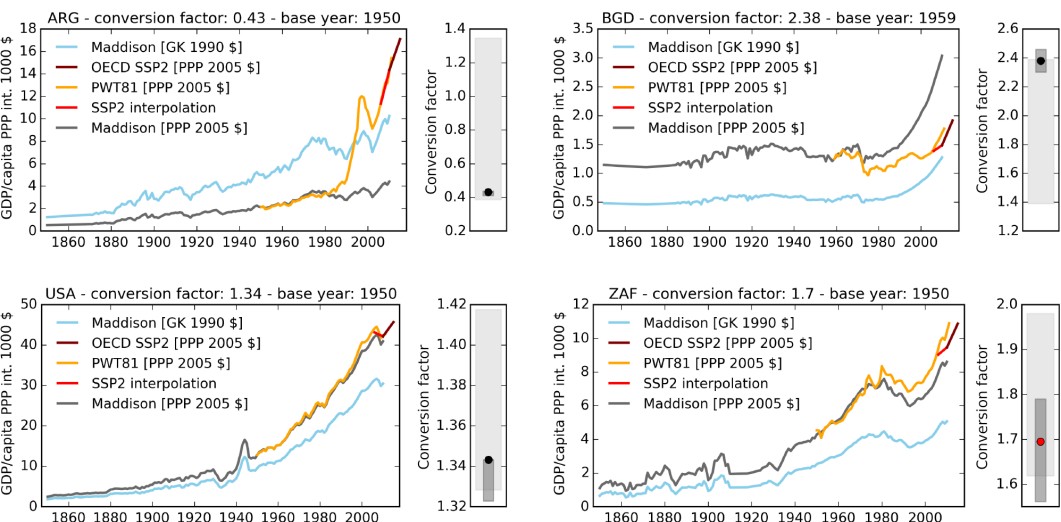

**Figure 4.** Results of matched GDP per capita time series for Maddison, PWT, and OECD SSP2 data for selected countries: ARG: Argentina; BDG: Bangladesh; USA: United States of America; ZAF: South Africa. Original Maddison data (blue, in 1990 GK USD) are matched to PWT data (yellow, in 2005 PPP USD) using a country-specific base year and conversion factor, to obtain converted Maddison data (gray, in 2005 PPP USD). Between 2006 and 2009, PWT data are interpolated (red) to match OECD SSP2 projections (maroon) starting in 2010. The sensitivity of the conversion factor over time is displayed to the right of each income time series. The black (red) point indicates whether the selected conversion factor was chosen based on the first overlapping year (5-year mean), while the dark gray and light gray shaded areas show the conversion factor min/max range for the first 5 overlapping years and for the 5-year running mean between the base year and 2005, respectively.

lated to match SSP population projections in 2010. For almost all countries the initial value of the SSP time series in 2010 and actually observed GDP values in 2010 only slightly deviate; see Figs. 4 and S1: only six countries show deviations larger than 10 % (AZE, Azerbaijan, GNQ, Equatorial Guinea, PNG, Papua New Guinea, SLV, El Salvador, TLS, Timor-Leste, ZWE, Zimbabwe). Even though the deviation illustrated in Fig. 4 corresponds to income deviations only, it also reflects GDP differences well as the SSP income projections are the main source of deviations, mostly due to the already mentioned fact that SSP simulations were completed before the financial crisis in 2008. However, Zimbabwe shows a large discrepancy of about 70 % that is caused by differences in both income and population estimates. When using Zimbabwe's time series for historical analysis only, we recommend truncation of the time series in 2005. Furthermore, for the following countries (Aruba, ABW, Antigua and Barbuda, ATG, Bermuda, BMU, Dominica, DMA, Federated States of Micronesia, FSM, Grenada, GRD, Kiribati, KIR, Saint Kitts and Nevis, KNA, Marshall Islands, MHL, Nauru, NRU, Seychelles, SYC, Tuvalu, TUV), no GDP projections exist in the OECD database. We therefore used the observational data from our reconstruction up to 2009.

For some small countries HYDE data have missing population values prior to certain years (Bermuda, BMU, Macao, MAC, Maldives, MDV < 1970; Federated States of Micronesia, FSM, Kiribati, KIR, Marshall Islands, MHL,

**Table 3.** Changes in 5-year running-mean conversion factors between the country-specific base year and 2005 for all 181 countries that have sufficient overlapping years of data. Where informative the specific countries are listed with their respective ISO3 codes; italicized countries lack individual country resolution in the MPD.

| Threshold in % | No. of countries | % of countries (181) | Selected list of countries |
|---|---|---|---|
| ≤ ±5 | 2 | 1.1 | HRV, UZB |
| ≤ ±10 | 14 | 7.7 | – |
| ≤ ±20 | 52 | 28.7 | – |
| ≤ ±25 | 69 | 38.1 | – |
| ≤ ±50 | 127 | 70.2 | – |
| ≤ ±100 | 156 | 86.2 | – |
| > ±100 | 25 | 13.8 | – |
| > ±200 | 10 | 5.5 | *ABW*, ARG, GAB, *GRD*, IRQ, *KNA*, KWT, *MAC*, *MDV*, QAT |

Nauru, NRU, French Polynesia, PYF, Seychelles, SYC, Tuvalu, TUV < 1960). Therefore, the corresponding GDP time series contains missing values for this period.

## 3 Data availability

We provide three different primary data sets, a data description file, and two supplementary data sets in the online archive at https://doi.org/10.5880/pik.2017.003.

1. A continuous table of global income data (in original 1990 G–K USD) based on the MPD for 160 individual countries and 3 groups of countries from 1850 to 2010.

2. A continuous and consistent table of global income data (in 2005 PPP USD) for 195 countries based on
the merged MPD and PWT8.1 data and extended using PWT9.0 and WDI data from 1850 to 2009, and consistent with OECD SSP2 income projections starting in 2010.

3. A continuous and consistent table of global GDP
data (in 2005 PPP USD) for 195 countries based on the merged MPD and PWT8.1 data, extended using PWT9.1 and WDI data, and consistent with OECD SSP2 GDP projections starting in 2010.

The second data set is illustrated in Fig. 5, showing the
20 population-weighted income distribution across continents and percentiles.

Furthermore, we provide two supplementary data sets.

1. A mask table complementing the first primary data set indicating which of the data points are original values
and which are estimated based on our present methodology, and used to generate Fig. 1.

2. A table of conversion factors (1990 GK USD to 2005 PPP USD) also indicating the data source used for matching and the methodology used for conversion, and
used to generate Fig. 2.

All data sets are provided in "csv" format and    freely available at https://doi.org/10.5880/pik.2017.003. Please also consult the data description file for additional information on the data set.

## 4 Discussion and conclusion

In conclusion, we provide a continuous time series for per capita income and GDP between 1850 and 2009 for 195 countries. The foundation of our work is the income data set maintained by the Maddison Project, which is completed
using various interpolation and extrapolation techniques and harmonized with economic data from the Penn World Tables. The main objective is to provide a continuous economic time series that is readily applicable across disciplines and by non-experts in the field. The methodology applied is rather simple
and comes with several limitations and caveats.

As we do not devise new historic economic figures ourselves, we are bound to existing data only that are combined with various techniques to replace missing values. Our

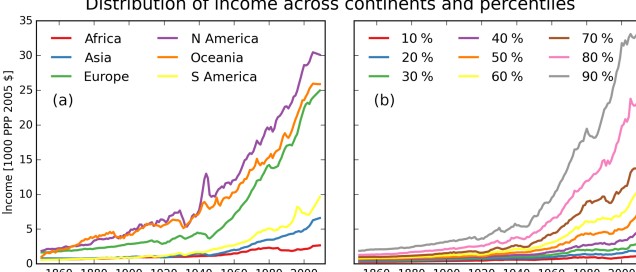

**Figure 5.** Distribution of population-weighted income over time across continents **(a)** and percentiles **(b)** that can be analyzed for, e.g., income inequality or other distributional effects. Percentiles rank the number of countries by average national income.

main assumption is that geographically close countries have observed similar growth paths over the last 150–200 years,
such that we can use neighboring (groups of) countries to estimate missing data. This assumption strongly depends on many political, economical, and societal aspects (e.g., the economic system, the membership in alliances, the occurrence of wars, the colonial background, and many more)
that, however, were not considered in our work. While rather exhaustive data exist for Western European countries, these limitations might be less of a problem than for most African countries. As a consequence, one should treat the data with care and allow for uncertainties, in particular where data cov-
erage is limited or almost non-existent.

Another limitation arises due to different units: the original MPD is measured in 1990 GK USD, while all other data sets use PPP equivalents in either 2005 or 2011 USD. The required PPP transformation can lead to underestimation or
65 overestimation of economic figures, in particular further back in time. As mentioned above for Brunei (discovery of fossil fuels) and Somalia (ethnic conflict and civil war), large conversion factors are due to rapid changes in a country's income over a short period of time that can deteriorate the PPP
conversion. However, as currently there exists no reliable solution to circumvent this conversion problem, one has to interpret transformed data with caution. For example, countries that mainly depend on fossil-fuel exports had rapid income jumps in the not so distant past that can overestimate their
income before oil discovery. For this reason we also provide the interpolated MPD in original 1990 GK USD. In addition to this, the choice of a specific base year for currency conversion introduces additional uncertainty about past estimates. For several reasons discussed above, we here systematically
rely on a conversion factor defined at the earliest possible time. While this provides a rather conservative estimate of historical income in general, a different base year for conversion can yield large differences for some countries, as our sensitivity assessment has shown.
Another problem arises due to shifting national borders, and the formation of new and disappearance of old nations:

the data sets only reflect the current political map defined by the list of countries available in the PWT8.1 database. To circumvent but not fully exclude this problem, we work with per capita income time series until the very last moment that are then multiplied by historical population data to generate GDP estimates. The income data are here provided as well such that the inclined user can generate GDP time series for a different political map himself. For consistency reasons population and income estimates are selected from identical sources, except for the MPD, where HYDE data are used. However, when merging different time series, inconsistencies arise due to different country definitions, e.g., as is the case for countries in the former Yugoslavia or for the transition between historical data and SSP projections. To reduce the inconsistencies to a minimum we rely on the country definitions in PWT8.1 and adjust the other sources to it. Regarding the former Yugoslavia, we split the population estimates for missing time periods for SCG (Serbia and Montenegro) into Serbian (SER) and Montenegrin (MNE) parts based on reported population ratios from 1990. Further, the population of Kosovo (KSV) was added to Serbia (SER) to match with the PWT's estimates. A similar procedure was followed for the respective population of Israel (ISR) and Palestine (PSE) prior to 1950 and 1969, respectively: reported population ratios were used to estimate a single country's population further back in time. No further adjustment was necessary for the other world regions.

Furthermore and as mentioned above, historical population estimates are unavailable for some small countries, making the GDP time series incomplete.

Despite these shortcomings and uncertainties, this new data set will broaden the applicability of historic estimates of economic activity, e.g., in the field of climate impact research in order to facilitate impact simulations on centennial timescales (Frieler et al., 2017). It further provides the opportunity to generate gridded GDP distributions for the past based on recent downscaling initiatives (Geiger et al., 2017; Murakami and Yamagata, 2017). Moreover, the increased research interest in past GDP data will provide valuable feedback to the historians and economists working in this field and might stimulate further advances. These advances are expected to improve this current data set further.

**The Supplement related to this article is available online at https://doi.org/10.5194/essd-10-1-2018-supplement.**

**Author contributions.** TG created and analyzed the data set and wrote the paper.

**Competing interests.** The author declares that he has no conflict of interest.

**Acknowledgements.** We wish to thank Johannes Gütschow for his valuable comments on the generation of this data set and Katja Frieler for her helpful remarks to improve the manuscript. We also thank Jutta Bolt for advice on the Maddison Project Database. We further thank Kirsten Elger from GFZ Data Services for invaluable support in creating the DOI data archive.

This work was funded through the framework of the Leibniz Competition (SAW-2013-PIK-5 and SAW-2016-PIK-1).

Edited by: David Carlson
Reviewed by: two anonymous referees

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

    worldbank.org/, last access: 11 January 2017.

## Remarks from the language copy-editor

CE1    "for"?
CE2    Do you perhaps mean AD 1 here?
CE3    This is how the name appears on the website. The image has also been changed accordingly.
CE4    Please confirm.
CE5    "data" is a plural noun, so its verb must be too.

## Remarks from the typesetter

TS1    Please note that if there is only a link in the text without a reference, the last access date has to be provided, e.g., last access: 1 March 2018.
TS2    Please confirm.
TS3    Please provide the short title.
TS4    The composition of all figures has been adjusted to our standards.
TS5    Please note that the "Data availability" title for this section is our journal standard.
TS6    Please provide the volume and article number.
TS7    Please confirm the insertion.
TS8    Please confirm the insertion.
TS9    Please check the DOI number.
TS10    Please provide the location of the publisher.
TS11    Please update if available.
TS12    Please provide the location of the publisher.