# Peer review of "Continuous national Gross Domestic Product (GDP) time series for 195 countries: past observations (1850-2005) harmonized with future projections according to the Shared Socio-economic Pathways (2006-2100)"

_Earth System Science Data, 2017_

## Referee Comment (RC1) · Anonymous Referee #1 · 25 Sep 2017

**1   General Comments**

The author presents a data set and methodology for harmonizing GDP trajectories across a broad historical time scale with projections provided by the Shared Socioeconomic Pathways. Historical data comes from a variety of sources, most notably the Penn World Tables and the Madison Project. The primary methodological advances

deal with converting currencies from units of GK1990 to PPP2005, the presentation of which needs further clarification. The resulting data set will be highly useful among the climate impact community when used in conjunction with other data sets (e.g., historical and projected population trajectories) to assess past and future impacts from climate change, natural hazards, etc.

**2  Specific Comments**

1. Please provide equations used for primary calculations, e.g., translating between currency units

2. A table summarizing Section 2 (which data set was used for which aspects of the analysis) should be provided

**2.1  Section 2.4**

1. When generating GK1990 to PPP2005 conversion factors, the author should explain why it is sufficient to develop factors only assessing the first 5-year period - overlapping data is available between the two data sets for much longer periods of time

2. Please show for a few representative countries time series of calculated conversion factors across the entire overlapping time series to show that the conversion factor is robust across time

3. Please report how many (and what kind) of countries required a mean conversion factor

4. The author should provide a (perhaps brief) analysis as to how sensitive are the results do the methodology used to generate conversion factors.

5. Figure 2b is entirely illegible; additionally, there is no presented logic as to why ISO codes are placed along the x-axis. Please rework this into a easier-to-digest representation of the main point of the figure. Please highlight in that figure outliers and countries of note with specific labels. The previous section breaks down the MPD into different regions which would be a reasonable approach here (e.g., different histograms for different regions). Additionally, if the figure is presented next to another figure (Figure 2a), the y-axis should have the same bounds.

6. If the 2008 financial crisis is the primary source of error between PWT and the SSP projections, why is 2005 taken as the base year of interpolation? Would it not be better to use 2008 in order to capture the dynamics of the financial crisis? Please provide a sound basis as to why 2005 is chosen as the base year for merging historical data with SSP data.

**3  Technical Corrections**

The final column in the posted datafiles is suffixed with ';;' which is interpreted as a string in most programming languages. This should be fixed so that all values are numeric when read in by an automated process

---

## Referee Comment (RC2) · Anonymous Referee #2 · 23 Jan 2018

The author presents a combination of existing datasets for GDP which have been standardised in various projects by various authors but never as far as I know in a journal publication, creating a useful dataset from 1850-2100 subject to the many assumptions made by the government. It is important to note firstly the limitations of using a combination of Penn World Tables, and Maddison values given the differing approaches often used to come up with their GDP values.

1) I would like to see the currency conversions made by the authors for each country, as the homogenous joining of datasets needs to be consistent in the periods where there are hyperinflation; large changes in currency, or differing values per year.

2) Datasets: There appears to be some inconsistencies in the final year of the files with the ;; reading in errors. Despite this, a sanity check of a few countries seems plausible.

The PPP conversions need to be better explained within the text and the assumptions made. The use of the SSPs is of course a key to why the PPPs were chosen in 2005, but why not use more than 1 year of PPP conversions, and what are the uncertainties?

I am missing also the context as to the quality of the overlaps between the various datasets, which likely needs to be explained for a reasonably constant country i.e. New Zealand, and a very volatile country (Zimbabwe).

Aruba (the first line) does not inspire me with confidence in this method - as it shows an 85% reduction in GDP in the period from 2005-2009, where Aruba was a reasonably constant country.

A sensitivity analysis is required for the input datasets and the uncertainties in the overlaps.

I also did not read much about border changes, and adjusting boundaries with respect to the GDP estimate adjustments, which should be alluded to and explained - "Existing discrepancies were harmonized and an interval between 2005 and 2010 was used to allow for a smooth transition to SSP projection" is not a reasonable explanation and details should be shown for the examples talked about (Yugoslavia, Colonial Africa etc.)

3) Quality of writing etc. The paper itself was coherently written, and although some of the figures are complicated and likely could be simplified for visual purposes (Fig 2), the writing is adequate.

- issues like GPD (should be GDP) in the abstract, and another check as to units should

be made.

All in all, this paper should be a useful addition to the journal, but needs some additional work to clear up some of the unknowns in the dataset combinations.

---

## Author Comment (AC1) · 20 Feb 2018

Reply to Referee 1

We thank the Referee for his appreciation of our work and his constructive and very useful comments. We think that the changes to the manuscript and the underlying dataset will improve the manuscript's quality and will meet the Referee's agreement.
Please find the Referee's comments (COM) and our reply (REP) with according changes to the manuscript below:

**General Comments**
COM: The primary methodological advances deal with converting currencies from units of GK1990 to PPP2005, the presentation of which needs further clarification.

REP: We have adapted and extended the manuscript at various instances to clarify the procedure and to inform about loopholes and uncertainties. We also created an appendix that now contains time series for each country considered and information about the country-specific conversion rates and their variability over time.

**Specific Comments**

1. COM: Please provide equations used for primary calculations, e.g., translating between currency units

REP: We have added equations at various instances to clarify our procedure.

2. COM: A table summarizing Section 2 (which data set was used for which aspects of the analysis) should be provided

REP: A table was added that makes it easier for the reader to understand how the different data sets interact and how they were used in our analysis.

**Specific Comments regarding Section 2.4**

1. COM: When generating GK1990 to PPP2005 conversion factors, the author should explain why it is sufficient to develop factors only assessing the first 5-year period - overlapping data is available between the two data sets for much longer periods of time

REP: We decided to develop the conversion schemes for the first 5 overlapping years of data for several reasons: 1) Data for several countries is not reported in the Maddison database and we estimate income trajectories for these countries based on the trajectories of country groups. Therefore, conversion to observed data at the earliest point in time is desirable; 2) As our final unit of merit is PPP 2005 USD, we chose to use original data with this unit as our main source and adjusted all other data sources to it; 3) For consistency (in line with point 1) and to avoid arbitrariness, we applied a systematic approach instead of a country-specific definition; 4) Using the earliest point in time also provides a lower bound for currency conversion for most countries (the all country-mean and median conversion factor grows over time, see also new figure in the manuscript) and thus avoids overestimation of past values, e.g. due to more recent high growth phases of certain countries; 5) We observe rather stable conversion factors over time for many countries (see the table for the sensitivity analysis in the revised manuscript.)

2. COM: Please show for a few representative countries time series of calculated conversion factors across the entire overlapping time series to show that the conversion factor is robust across time

REP: In the appendix to the revised manuscript one can now find time series for each country considered and information about the country-specific conversion rates and their variability over time. We also updated Figure 3 using this newly available information and added a new paragraph discussing the robustness and sensitivity of our methodology.

3. COM: Please report how many (and what kind) of countries required a mean conversion factor

REP: We added the required information to the manuscript, as it was formerly only available in the DOI archive. It now also appears in the appendix of the manuscript for each country specifically.

4. COM: The author should provide a (perhaps brief) analysis as to how sensitive are the results do the methodology used to generate conversion factors.

REP: We added a paragraph discussing the robustness and sensitivity, see also our reply to comment 2 above.

5. COM: Figure 2b is entirely illegible; additionally, there is no presented logic as to why ISO codes are placed along the x-axis. Please rework this into a easier-to-digest representation of the main point of the figure. Please highlight in that figure outliers and countries of note with specific labels. The previous section breaks down the MPD into different regions which would be a reasonable approach here (e.g., different histograms for different regions). Additionally, if the figure is presented next to another figure (Figure 2a), the y-axis should have the same bounds.

REP: We generated a new figure according to the Referee's suggestion as well as adapted the surrounding discussion in the manuscript. The new figure now also shows the variability in conversion rates for different world regions.

6. COM: If the 2008 financial crisis is the primary source of error between PWT and the SSP projections, why is 2005 taken as the base year of interpolation? Would it not be better to use 2008 in order to capture the dynamics of the financial crisis? Please provide a sound basis as to why 2005 is chosen as the base year for merging historical data with SSP data.

REP: The database was primarily designed to be usable in combination with the SSP projections, which are available from 2010 onwards and were generated many years before the financial crisis. Thus, the unexpected occurrence of the financial crisis but also other unforeseen events and uncertainties in projected growth patterns have contributed to a mismatch of observed and SPP-projected GDP values in 2010. Although this seems outdated, this is required for consistency reasons: all other data sources in the field of climate impact research with which the present database might be matched are based on the 2010 SSP projections as well. If we had chosen 2008 or even 2009 (when most countries experienced the full impact of the financial crisis) as a basis for interpolation, we

would have only added 3-4 years based on observations but would have produced large and (and even more) unrealistic kinks in most countries' time series thereafter.

**Technical comments**

COM: The final column in the posted datafiles is suffixed with ';;' which is interpreted as a string in most programming languages. This should be fixed so that all values are numeric when read in by an automated process

REP: The ";;" symbols in the final column were wrongly added when finalizing the data and overlooked at the final check. The updated version of the DOI source will be corrected for this error.

---

## Author Comment (AC2) · 20 Feb 2018

Reply to Referee 2

We thank the Referee for his appreciation of our work and his constructive and very useful comments. We think that the changes to the manuscript and the underlying dataset will improve the manuscript's quality and will meet the Referee's agreement.
Please find the Referee's comments (COM) and our reply (REP) including the according changes to the manuscript below:

**Specific Comments**

1) COM: I would like to see the currency conversions made by the authors for each country, as the homogenous joining of datasets needs to be consistent in the periods where there are hyperinflation; large changes in currency, or differing values per year.

REP: We created an appendix that now contains time series for each country considered and information about the country-specific conversion rates and their variability over time. We also added a paragraph in the manuscript discussing the sensitivity of our findings.

2) COM: Datasets: There appears to be some inconsistencies in the final year of the files withthe ;; reading in errors. Despite this, a sanity check of a few countries seems plausible.

REP: The ";;" symbols in the final column were wrongly added when finalizing the data and overlooked at the final check. The updated version of the DOI source will be corrected for this error.

COM cont'd: The PPP conversions need to be better explained within the text and the assumptions made. The use of the SSPs is of course a key to why the PPPs were chosen in 2005, but why not use more than 1 year of PPP conversions, and what are the uncertainties? I am missing also the context as to the quality of the overlaps between the various datasets, which likely needs to be explained for a reasonably constant country i.e. New Zealand, and a very volatile country (Zimbabwe).

REP: We adapted and extended the manuscript at various instances to clarify the procedure and to inform about uncertainties of choices we have made when creating the dataset. We also created an appendix that now contains time series for each country considered and information about the country-specific conversion rates and their variability over time. Based on this new information, we discuss the sensitivity of our currency conversion methodology.
As noted by the Referee, the current dataset was primarily designed to be usable in combination with the SSP projections and thus uses the 2005 PPPs only. Creating a comparable dataset for different PPPs would be desirable but is beyond the scope of this current exercise that we leave to future work.

COM cont'd: Aruba (the first line) does not inspire me with confidence in this method - as it shows an 85% reduction in GDP in the period from 2005-2009, where Aruba was a reasonably constant country. A sensitivity analysis is required for the input datasets and the uncertainties in the overlaps.

REP: As stated in the original manuscript, Aruba is the largest outlier in the full sample when it comes to matching with the SSPs. After double-checking the original OECD SSP2 data

source, that was used for matching between 2005 and 2010, we convinced ourselves that the GDP of Aruba is not correctly stated there. As Aruba is not officially listed within IIASA's SSP archive (https://secure.iiasa.ac.at/web-apps/ene/SspDb/dsd?Action=htmlpage&page=welcome) (and we did not receive feedback from the OECD on their source), we therefore decided not to include SSP data for Aruba and provide only observational data until 2009 (as we also did for countries like Antigua and Barbuda (ATG), Bermuda (BMU), Dominica (DMA), Federated States of Micronesia (FSM), Grenada (GRD), Kiribati (KIR), Saint Kitts and Nevis (KNA), Marshall Islands (MHL), Nauru (NRU), Seychelles (SYC), Tuvalu (TUV), see manuscript for details).

We also checked other potential candidates for faulty data (e.g. Zimbabwe) but could not detect any other issues.

COM cont'd: I also did not read much about border changes, and adjusting boundaries with respect to the GDP estimate adjustments, which should be alluded to and explained - "Existing discrepancies were harmonized and an interval between 2005 and 2010 was used to allow for a smooth transition to SSP projection" is not a reasonable explanation and details should be shown for the examples talked about (Yugoslavia, Colonial Africa etc.)

REP: Most of the occurring border changes are circumvented (but not fully excluded) through the use of GDP per capita instead of nominal GDP. Nominal GDP is calculated in the end from country-level GDP per capita and population data. To do so, we rely on the list of countries and country definitions as used in the PWT v8.1 in their current state. To make the calculation of GDP feasible, we adjusted, in particular, population data to match country-specific income data. Adjustments mostly affect former Yugoslavia and Israel/Palestine. As historical data is very scarce for the African continent and estimates are rather uncertain (see also the discussion in the manuscript) we only used the country borders / country names in their current state without further adjustments. Also note that "new" countries like South Sudan are not included in our database.

The revised manuscript now contains a more detailed discussion on the harmonization of data sources and the underlying assumptions.

3) COM: Quality of writing etc. The paper itself was coherently written, and although some of the figures are complicated and likely could be simplified for visual purposes (Fig 2), the writing is adequate. - issues like GPD (should be GDP) in the abstract, and another check as to units should be made.

REP: Figure 2 was completely revised and provides now a comparison of conversion factors for different world region, as suggested by the other Referee. Further, the typo in the abstract was corrected, and all units were double-checked.